**Brief Communication**

# Deep-learning electronic-structure calculation of magnetic superstructures

He Li [1,2,3,7], Zechen Tang[1,7], Xiaoxun Gong[1,4], Nianlong Zou [1], Wenhui Duan [1,2,3,5] ✉ & Yong Xu [1,2,5,6] ✉

Ab initio studies of magnetic superstructures are indispensable to research on emergent quantum materials, but are currently bottlenecked by the formidable computational cost. Here, to break this bottleneck, we have developed a deep equivariant neural network framework to represent the density functional theory Hamiltonian of magnetic materials for efficient electronic-structure calculation. A neural network architecture incorporating a priori knowledge of fundamental physical principles, especially the nearsightedness principle and the equivariance requirements of Euclidean and time-reversal symmetries ($E(3) \times \{I, \mathcal{T}\}$), is designed, which is critical to capture the subtle magnetic effects. Systematic experiments on spin-spiral, nanotube and moiré magnets were performed, making the challenging study of magnetic skyrmions feasible.

The subject of magnetic superstructures, such as magnetic skyrmions, moiré magnetism and spin-spiral magnets, has attracted intensive research interest, opening opportunities to explore emergent physics in quantum materials, including the skyrmion Hall effect, the topological Hall effect, flat-band physics and so on[1]. Ab initio calculations based on density functional theory (DFT) has become an indispensable tool for research, but is only applicable to the study of small-scale superstructures owing to the high computational cost. The recent development of deep learning ab initio methods[2–15] has shed light on solving this bottleneck problem; these methods use artificial neural networks to learn from ab initio data and apply neural networks for material simulation without invoking ab initio codes, enabling the study of large-scale material systems. However, current methods are usually designed to treat systems without magnetism, which neglects the dependence of material properties on magnetic structure; thus, they are not suitable for the purpose.

A key challenge of deep learning DFT calculations is to design deep neural network models to represent the DFT Hamiltonian $H_{DFT}$ for efficient electronic-structure simulation[8,9]. This problem has recently been investigated for non-magnetic systems[7–13]. The counterpart problem for magnetic systems is of great importance but faces some critical challenges. First, an extra dependence on magnetic structure $\{\mathcal{M}\}$

is introduced into $H_{DFT}$ (Fig. 1a), which is physically distinct from the dependence on atomic structure $\{\mathcal{R}\}$. Second, the spin degrees of freedom are usually negligible in the non-magnetic case, but become essential for magnetic systems. Consequently, $H_{DFT}$ becomes non-diagonal in spin space, leading to a larger number of non-zero matrix elements (Fig. 1a). Third, satisfying fundamental symmetry conditions is a prerequisite for achieving good performance in deep learning problems[8–11]. Generalized symmetry requirements for neural network models of $H_{DFT}$ are imposed by symmetry operations on both $\{\mathcal{M}\}$ and $\{\mathcal{R}\}$. This important issue has not been addressed before. Last but not least, high-precision calculations are required to describe the subtle magnetic effects, setting high standards for method development. In this context, substantial generalization of deep learning DFT methods is urgently demanded by the research field.

In this work, we develop an extended deep learning DFT Hamiltonian (xDeepH) method, including theoretical framework, numerical algorithm and computational code, which learns the dependence of the spin–orbital DFT Hamiltonian on atomic and magnetic structures by deep equivariant neural network (ENN) models, enabling efficient electronic-structure calculation of large-scale magnetic materials. As a critical innovation, we design an ENN architecture to incorporate physical insights and respect the fundamental symmetry group $E(3) \times \{I, \mathcal{T}\}$

[1]State Key Laboratory of Low Dimensional Quantum Physics and Department of Physics, Tsinghua University, Beijing, China. [2]Tencent Quantum Laboratory, Tencent, Shenzhen, China. [3]Institute for Advanced Study, Tsinghua University, Beijing, China. [4]School of Physics, Peking University, Beijing, China. [5]Frontier Science Center for Quantum Information, Beijing, China. [6]RIKEN Center for Emergent Matter Science (CEMS), Wako, Japan. [7]These authors contributed equally: He Li, Zechen Tang. ✉e-mail: duanw@tsinghua.edu.cn; yongxu@mail.tsinghua.edu.cn

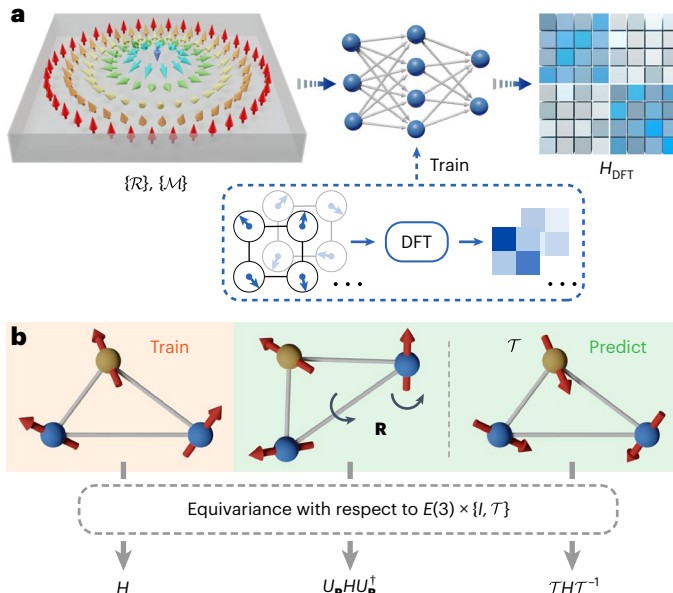

**Fig. 1 | Extended deep learning DFT Hamiltonian (xDeepH) method for studying magnetic materials. a**, Workflow of xDeepH. Equivariant neural networks (middle) are used to represent the DFT Hamiltonian $H_{DFT}$ (right) as a function of atomic structure $\{\mathcal{R}\}$ and magnetic structure $\{\mathcal{M}\}$ (left). The neural network models are trained on DFT data on small-size structures (blue dashed box) and applied to study magnetic superstructures, such as magnetic skyrmions. The colored arrows denote magnetic moments. **b**, Equivariance of $H_{DFT}$ ($\{\mathcal{R}\}, \{\mathcal{M}\}$) with respect to the $E(3) \times \{I, \mathcal{T}\}$ group, where $E(3)$ is the Euclidean group in three-dimensional space, $I$ is the identity operator and $\mathcal{T}$ is the time-reversal operator. Transformations of rotation **R** and time-reversal $\mathcal{T}$ are illustrated. The colored balls and arrows denote atoms and magnetic moments, respectively.

(Euclidean and time-reversal symmetries) in the representation of $H_{DFT}$ ($\{\mathcal{R}\}, \{\mathcal{M}\}$), ensuring efficient and accurate deep learning. $E(3)$ is the Euclidean group in three-dimensional space, $I$ is the identity operator and $\mathcal{T}$ is the time-reversal operator. The method is systematically tested to show high precision (submillielectronvolt error) and good transferability by example studies of magnetic superstructures ranging from spin-spiral, nanotube and moiré magnets to magnetic skyrmions. Benefiting from extended capability and state-of-the-art performance, xDeepH could find promising applications in future materials research and stimulate the development of deep learning ab initio methods.

## Results and discussion
### Theoretical framework of xDeepH
The deep learning DFT Hamiltonian (DeepH) method has been developed to improve the efficiency of electronic-structure calculation, which shows great potential to address the accuracy-efficiency dilemma of DFT[8,9]. A substantial generalization of the method is required to study a broad class of magnetic materials. For non-magnetic systems, $H_{DFT}$ as a function of atomic structure $\{\mathcal{R}\}$ is calculated by self-consistent field (SCF) iterations in DFT. The function $H_{DFT}$ ($\{\mathcal{R}\}$) is the learning target of DeepH. In contrast, for magnetic systems, $H_{DFT}$ depends not only on atomic structure but also on magnetic structure $\{\mathcal{M}\}$. To compute $H_{DFT}$ for a given $\{\mathcal{M}\}$, one needs to apply constrained DFT that employs the Lagrangian approach to constrain magnetic configuration and introduce constraining fields into the Kohn–Sham potential[16]. In general, the mapping from $\{\mathcal{R}\}$ and $\{\mathcal{M}\}$ to the spin–orbital $H_{DFT}$ is unique in constrained DFT[17]. $H_{DFT}$ for magnetic systems is also calculated by SCF iterations but requires much more computational resource than the non-magnetic counterpart. This is because the additional constraining fields should be determined self-consistently, and an enlarged Hamiltonian matrix non-diagonal in spin space must be considered.

The workflow of xDeepH is illustrated in Fig. 1a. First, small-size materials (simulated by small supercells) with diverse atomic and magnetic configurations are calculated by constrained DFT for preparing datasets. Then deep neural networks representing $H_{DFT}$ ($\{\mathcal{R}\}, \{\mathcal{M}\}$) are trained on the datasets. Next, the neural networks are applied to predict $H_{DFT}$ for materials with varying atomic and magnetic structures. Based on $H_{DFT}$, any electronic properties of materials in the single-particle picture can be computed. By replacing the DFT SCF calculation with deep neural networks, the method greatly reduces the computational cost of electronic-structure calculation and enables the study of magnetic superstructures (for example, magnetic skyrmions). The critical issue, however, is the design of intelligent neural networks for modeling the mapping function ($\{\mathcal{R}\}, \{\mathcal{M}\}) \mapsto H_{DFT}$, using as much a priori knowledge as possible for optimizing neural network performance.

Two physical principles are essential to the deep learning problem here, including the nearsightedness (or locality) principle of electronic matter[18] and the symmetry principle. Local physical properties satisfying the nearsightedness principle are insensitive to distant change of the chemical environment[18]. For instance, charge density is a local property, whereas the Kohn–Sham eigenstates are non-local. The latter depends sensitively on the global material structure, which is complicated from the point of view of machine learning. In general, qualities with local properties are more favorable for deep learning than non-local ones[19–21]. Moreover, the fundamental physical equations are equivariant under symmetry operations (for example, translation, rotation and so on). The symmetry is an inherent property of physical data. Hence the use of symmetry properties could substantially facilitate deep learning. In short, the principles of locality and symmetry are a priori knowledge of pivotal importance to artificial intelligence.

Let us first check the locality nature of the deep learning target $H_{DFT}$ ($\{\mathcal{R}\}, \{\mathcal{M}\}$). In DFT calculations, plane waves and localized orbitals are usually employed as basis functions. The localized basis is preferred, as it is compatible with the locality principle. Under the localized basis, the DFT Hamiltonian can be viewed as an ab initio tight-binding Hamiltonian (see details in 'DFT Hamiltonian under localized basis' in Methods). The Hamiltonian matrix block $H_{ij}$, which describes hopping between atoms $i$ and $j$, has vanishing values for atom pairs with distance $d_{ij} > R_C$. The cut-off radius $R_C$ is determined by the spread of orbital functions and usually on the order of a few ångströms. Importantly, $H_{ij}$ ($\{\mathcal{R}\}, \{\mathcal{M}\}$) obeys the nearsightedness principle, which is influenced only by changes in the chemical environment of finite range $R_N$.

Noticeably, the influence induced by changes in $\{\mathcal{R}\}$ and $\{\mathcal{M}\}$ are relevant to two kinds of nearsightedness length scales, denoted as $R_{N1}$ and $R_{N2}$, respectively. Formally, varying $\{\mathcal{R}\}$ will alter the strong external potential in $H_{DFT}$, whereas varying $\{\mathcal{M}\}$ will mainly modify the relatively weak constraining fields, leading to minor influence on $H_{DFT}$. It is thus expected that the latter influence on $H_{DFT}$ is weaker in magnitude and shorter in length scale ($R_{N2} < R_{N1}$). Our results suggest that $R_{N2} \approx R_C$ (Supplementary Fig. 1), and $R_{N1}$ is typically several times larger than $R_C$. The two distinct dependence behaviors of $H_{DFT}$ ($\{\mathcal{R}\}, \{\mathcal{M}\}$) should be accurately described together by the deep learning method. This important issue will be addressed in the dataset preparation and neural network design.

The symmetry principle is another a priori knowledge of essential importance to optimize deep learning performance. On $H_{DFT}$ ($\{\mathcal{R}\}, \{\mathcal{M}\}$), the fundamental symmetry group is $E(3) \times \{I, \mathcal{T}\}$, which includes the $E(3)$ group (including translation, rotation and spatial inversion) in direct product with identity $I$ and time reversal $\mathcal{T}$. These symmetry operations may act on the atomic and magnetic structures or on both orbital and spin spaces of $H_{DFT}$, leading to equivariant transformation requirements, as illustrated in Fig. 1b for spatial rotation and time reversal. With information about one structure, $H_{DFT}$ of all the

symmetry-related structures can be predicted via equivariant transformations. Thus, the benefit of incorporating a symmetry principle into deep learning is considerable. The equivariance transformations of the DFT Hamiltonian required by the $E(3) \times \{I, \mathcal{T}\}$ symmetry group are described in 'Equivariance transformations of DFT Hamiltonian' in Methods.

The Euclidean symmetry requirements can be preserved using the framework of ENNs[22,23]. All of the feature vectors in ENNs have the equivariant form $\mathbf{x}_m^l$, which carries the irreducible representation of the SO(3) group of dimension $2l + 1$, where $l$ is an integer and $m$ is an integer or half-integer varying between $-l$ and $l$. Feature vectors can be converted to equivariant tensors (or vice versa) to construct the DFT Hamiltonian via the Wigner–Eckart theorem: $l_1 \otimes l_2 = |l_1 - l_2| \oplus \cdots \oplus (l_1 + l_2)$, which changes tensor product '$\otimes$' into direct sum '$\oplus$'. How to realize Euclidean equivariance is described in 'Realization of Euclidean equivariance' in Methods.

However, it seems difficult to handle the time-reversal equivariance within the original ENN framework, because time reversal introduces a non-trivial transformation of the DFT Hamiltonian in the spin space (equation (2) in Methods), and complex-valued quantities are prevalent in the problem. We find that this problem is solved by applying the transformation $\frac{1}{2} \otimes \frac{1}{2} = 0 \oplus 1$. Details of the time-reversal symmetry are described in Supplementary Section 2. Under the $0 \oplus 1$ representation, the effect of the time-reversal operator becomes very simple: the signs are flipped for the imaginary part of the $l = 0$ vector and the real part of the $l = 1$ vector, and all others are unchanged. This means, if we want to introduce time-reversal symmetry to the ENN, we need to let the ENN output the real and imaginary parts of the $l = 0$ and $l = 1$ vectors separately and introduce an additional index $t$ ($t = 0, 1$) into the rotation-equivariant vectors to mark their 'time-reversal parity': $\mathbf{x}_m^{l,t} \xrightarrow{\mathcal{T}} (-1)^t \mathbf{x}_m^{l,t}$. This additional parity should be taken care of for all the input, output and internal vectors throughout the ENN, but it can be treated in exactly the same way as the spatial inversion. Spatial-inversion symmetry is already implemented in our ENN framework, so introducing another parity index is a straightforward generalization. Therefore, all the symmetry requirements by the $E(3) \times \{I, \mathcal{T}\}$ group can be obeyed by the generalized ENN framework.

Here we describe the ENN architecture for xDeepH. xDeepH is based on a message-passing neural network[4], using graphs of vertices and edges to represent materials. Each atom associated with nuclear charge $Z_i$ and magnetic moment $\mathbf{m}_i$ is denoted by a vertex. Atomic pairs with non-zero $H_{ij}$ are connected by directional edges associated with interatomic distance vector $\mathbf{r}_{ij}$. These features of vertices and edges are input to an embedding layer, which is used to construct initial equivariant feature vectors of different $l$. Each vector is also labeled by the spatial-inversion and time-reversal parities ($p, t$). For instance, $\mathbf{r}_{ij}$ has ($p = 1, t = 0$), and $\mathbf{m}_i$ has ($p = 0, t = 1$). The ENN iteratively updates the equivariant features for vertices $\mathbf{v}_i$ and edges $\mathbf{e}_{ij}$ by updating them using features of their neighborhood as proposed in DeepH-E3[11]. Note that non-linear activation functions are allowed for $l = 0$ vectors only, and only even or odd activation functions are allowed to act on vectors with $p = 1$ or $t = 1$. Otherwise, the equivariance of the ENN will get broken. Importantly, interaction between vectors of different $l$ is implemented using a tensor product layer[22], $\mathbf{z}_{m_3}^{l_3} = \sum_{m_1, m_2} C_{m_1 m_2 m_3}^{l_1 l_2 l_3} \mathbf{x}_{m_1}^{l_1} \mathbf{y}_{m_2}^{l_2}$, that improves the capacity of ENN, where $C_{m_1 m_2 m_3}^{l_1 l_2 l_3}$ are Clebsch-Gordan coefficients and $\mathbf{x}, \mathbf{y}, \mathbf{z}$ are equivariant vectors. Moreover, as the feature vectors are translation invariant, the translation invariance of ENN is respected. Contributions from different atoms are aggregated by summation such that the updated features are invariant with respect to atomic permutations.

An overview of the neural network architecture of xDeepH is presented in Fig. 2. xDeepH embeds atomic and magnetic structures as initial vertex and edge features, followed by successive vertex layers and edge layers to update corresponding features. Distant information of atomic structure $\{\mathcal{R}\}$ is aggregated into features upon successive stacking of layers. The influence of the local magnetic moment on the DFT Hamiltonian is more localized. Regarding this locality, we introduce magnetic information $\{\mathcal{M}\}$ to the message-passing neural network with strict locality, as illustrated in the magnetic moment layer in Fig. 2b. We find that the approach makes training more efficient and accurate compared with treating $\{\mathcal{R}\}$ and $\{\mathcal{M}\}$ on an equal footing (see ablation studies in Supplementary Table 1). Finally, equivariant features on edges $\mathbf{e}_{ij}$ are used to construct $H_{ij}$. The xDeepH model is trained using DFT datasets by minimizing the loss function defined as the mean-squared errors of the DFT Hamiltonian matrix elements. Constraint DFT calculations are performed by using the OpenMX package[24]. DFT datasets are prepared by calculating many magnetic configurations for any given atomic structure, which increases the learning weight on the subtle magnetic effects. More details are described in 'Dataset preparation' in Methods. A previous study developed the deep neural network SpookyNet to learn the total energy and atomic forces of molecules, which not only considers the nuclear degrees of freedom as input but also takes the electronic degrees of freedom (including the total charge and spin) into account[6]. It is also based on the ENN method. A comprehensive comparison between xDeepH and SpookyNet is presented in Supplementary Section 7.

## Capability of xDeepH

The performance of xDeepH is tested by example studies of increasing complexity. The first case study is on monolayer $NiBr_2$, in which spin-spiral magnetism exists[25]. We prepare DFT datasets by calculating supercells of monolayer $NiBr_2$ with equilibrium lattice structure and random magnetic moment orientations at Ni sites (Extended Data Fig. 1a), and use them for training, validation and testing. The distribution of the mean absolute error (MAE; defined for the DFT Hamiltonian matrix elements, if not mentioned explicitly) for the test set is shown in Supplementary Fig. 2. The average MAE is as low as 0.56 meV. This ensures reliable predictions of band structures for changing magnetic configurations (Supplementary Fig. 3). Then xDeepH is applied to study the band structure of the $19 \times 1$ spiral magnetism and obtains results that are well consistent with DFT (Extended Data Fig. 1b), demonstrating the good generalization ability of the method.

The second case study is on nanotubes of monolayer $CrI_3$, which is one of the most investigated two-dimensional magnetic materials[26]. Recent research on $CrI_3$ nanotubes has revealed an intriguing magnetic transition from ferromagnetism to curved magnetism with increasing nanotube diameter[27]. The corresponding electronic-structure study, however, is lacking. We prepare DFT datasets by calculating flat monolayer sheets of $CrI_3$ with randomly perturbed atomic and magnetic configurations, and apply the trained xDeepH for the research (Extended Data Fig. 1c). On the test sets of monolayer sheets, the averaged MAE is down to 0.36 meV, and the band structures predicted by xDeepH match well with DFT results, showing quite high prediction accuracy (Supplementary Figs. 5 and 6). Test calculations are performed to compare the computational cost between xDeepH and DFT. The results indicate that the efficiency advantage of xDeepH over DFT is substantial and becomes more considerable as the system size increases (Supplementary Section 5). We further use xDeepH to study (16, 16) $CrI_3$ nanotubes for the two magnetic configurations proposed by ref. 27. As the curved lattice and magnetism are unseen during the training process, this is a strict test for xDeepH, especially on its equivariant performance. Remarkably, the predicted band structure (Extended Data Fig. 1d) and electric susceptibility (Supplementary Fig. 7) are in perfect agreement with DFT benchmark data. The variation in magnetism has a subtle influence on the band structure (Supplementary Fig. 8), and the subtle magnetic effects are well captured by xDeepH. Therefore, this method is promising for studying magnetic superstructures.

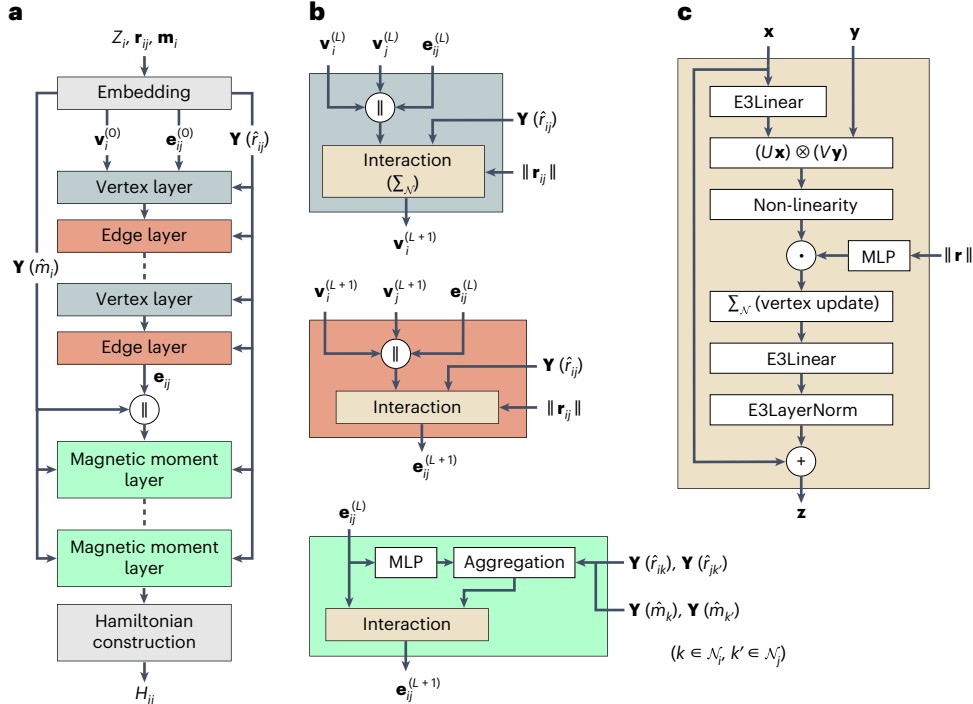

**Fig. 2 | Neural network architecture of xDeepH. a**, Workflow of the xDeepH model. Initial vertex and edge features are embedded by one-hot encoding and Gaussian expansion, respectively. Features are updated alternately by vertex layer and edge layer with interatomic distance vectors $\mathbf{r}_{ij}$ equipped with spherical harmonics $\mathbf{Y}_{lm}$. Subsequently, a magnetic moment layer with strict locality is used to include the magnetic moments $\mathbf{m}_i$ of atoms as input. Finally, a Hamiltonian construction layer is employed to build the DFT Hamiltonian matrix block $H_{ij}$. **b**, Details of the vertex layer (top), edge layer (middle) and magnetic moment layer (bottom), containing the 'Interaction' block. **c**, Details of the 'Interaction' block. The superscript ($L$) refers to the $L$th layer. || denotes vector concatenation and · denotes element-wise multiplication. $\sum_{\mathcal{N}}$ denotes the summation of neighbors for features, which is only valid for the vertex layer.

Finally, we try a challenging study on quantum materials with magnetic skyrmions. For this, we choose the moiré-twisted bilayer $CrI_3$ (Extended Data Fig. 2a), which has recently been predicted to have magnetic skyrmions originating from the stacking-dependent interlayer magnetic coupling of $CrI_3$ (ref. 28). The influence of magnetic skyrmions on electronic properties has rarely been studied before. The neural network models are trained on DFT datasets obtained for sample structures with a fixed twist angle $\theta = 60°$, and then applied to study moiré-twisted structures of varying $\theta$. Band structures and density of states obtained from xDeepH match well with the DFT results for a new twist angle $\theta = 81.79°$ (Supplementary Fig. 9), verifying the reliability of the approach. Next, we consider a twist angle of $\theta = 63.48°$, which has magnetic skyrmions in a large supercell[28], as illustrated in Extended Data Fig. 2a. For the ferromagnetic configuration, an extremely flat valence band emerges in the system (Extended Data Fig. 2b), which originates from the moiré twist. Noticeably, the flat band disappears in the presence of magnetic skyrmions (Extended Data Fig. 2c), indicating strong coupling between the two entities. This intriguing physics relevant to flat bands and magnetic skyrmions will be further explored in our future work.

## Discussion

The xDeepH method enables efficient electronic-structure calculations of large-scale magnetic materials, opening opportunities to explore emergent quantum physics and diverse magnetic systems, such as magnetic alloys, collinear or non-collinear magnetic systems, and periodic or non-periodic magnets. Moreover, the theoretical framework of xDeepH could find general applications in future development of deep learning ab initio methods, including the investigation of many-body interactions. For instance, xDeepH can be adapted to design neural networks in full respect of fundamental physical principles for representing other physical quantities, which is useful for exploring general physical effects relevant to spin dynamics, electron–magnon and phonon–magnon coupling, and so on. A more detailed discussion is included in Supplementary Section 9. However, the current xDeepH approach is not without limitations. For instance, considerable time is required to perform spin-constrained DFT calculations and for training neural network models, which hinders the dataset preparation and neural network training, respectively. In-depth research is thus needed to develop advanced spin-constrained DFT algorithms and to optimize the neural network architecture.

## Methods

### DFT Hamiltonian under localized basis

Localized basis is compatible with the locality principle. The orbital functions have the form $\phi_{iplm}(\mathbf{r}) = R_{ipl}(r)Y_{lm}(\hat{r})$, where the radial function $R_{ipl}(r)$ is centered at the $i$th atom, $p$ is the multiplicity index, and the angular part $Y_{lm}(\hat{r})$ is the spherical harmonics of degree $l$ and order $m$. Herein the spin degree of freedom must be considered, which is labeled by $\sigma = \pm 1/2$. The matrix element is then written as $[H_{ij}]_{m_1\sigma_1,m_2\sigma_2}^{p_1p_2;l_1l_2}$, where the subscript 'DFT' is omitted for simplicity. $H_{ij}$ is the Hamiltonian matrix block describing hopping between atoms $i$ and $j$. A notation of Hamiltonian matrix subblock $\mathbf{h} \equiv [H_{ij}]^{p_1p_2}$ is introduced, the elements of which have the form $\mathbf{h}_{m_1\sigma_1,m_2\sigma_2}^{l_1l_2}$.

### Equivariance transformations of DFT Hamiltonian

To study magnetic systems, one must consider an additional input of $\{\mathcal{M}\}$ and take the spin space of the electronic Hamiltonian into account. The Hamiltonian matrix subblock is the basic transformation unit of symmetry operations. For instance, if applying an rotation $\mathbf{R}$ on $\{\mathcal{R}\}$ and $\{\mathcal{M}\}$, the Hamiltonian matrix subblock $\mathbf{h}$ will transform as follows:

$$\mathbf{h}^{l_1 l_2}_{m_1 \sigma_1, m_2 \sigma_2} \xrightarrow{\mathbf{R}}$$

$$\sum_{m_1', \sigma_1', m_2', \sigma_2'} D^{l_1 \otimes \frac{1}{2}}_{m_1 \sigma_1, m_1' \sigma_1'}(\mathbf{R}) \left[ D^{l_2 \otimes \frac{1}{2}}_{m_2 \sigma_2, m_2' \sigma_2'}(\mathbf{R}) \right]^* \mathbf{h}^{l_1 l_2}_{m_1' \sigma_1', m_2' \sigma_2'}, \tag{1}$$

where the superscript $*$ denotes the complex conjugate, and $D^{l_1 \otimes \frac{1}{2}}(\mathbf{R})$ is the direct product of Wigner $D$-matrices $D^l(\mathbf{R})$ and $D^{\frac{1}{2}}(\mathbf{R})$, which act on the orbital and spin spaces, respectively. Under $\mathcal{T}$, $\{\mathcal{M}\}$ changes sign, $\{\mathcal{R}\}$ is unchanged and $H_{\text{DFT}}$ is transformed by an antiunitary operator. As real-valued orbital functions are employed here, the time-reversal transformation gives

$$\mathbf{h}^{l_1 l_2}_{m_1 \sigma_1, m_2 \sigma_2} \xrightarrow{\mathcal{T}} \begin{cases} \left( \mathbf{h}^{l_1 l_2}_{m_1(-\sigma_1), m_2(-\sigma_2)} \right)^* & \sigma_1 = \sigma_2 \\ -\left( \mathbf{h}^{l_1 l_2}_{m_1(-\sigma_1), m_2(-\sigma_2)} \right)^* & \sigma_1 \neq \sigma_2. \end{cases} \tag{2}$$

On spatial inversion, $\{\mathcal{R}\}$ and $\{\mathcal{M}\}$ have odd and even parities, respectively. $\mathbf{h}^{l_1 l_2}$ also has a well-defined parity, which is $(-1)^{l_1 + l_2}$. Moreover, $H_{\text{DFT}}(\{\mathcal{R}\} - \mathbf{r}_0, \{\mathcal{M}\})$ is invariant under spatial translation, where $\mathbf{r}_0$ denotes the origin and only the relative atomic positions are relevant to the problem. Besides, $H_{\text{DFT}}$ is equivariant under atomic permutation.

## Realization of Euclidean equivariance

To maintain Euclidean equivariance, all the input, internal and output feature vectors of ENNs are required to be equivariant. If a rotation $\mathbf{R}$ is applied to the input coordinates of ENNs, the equivariant feature vectors will transform accordingly: $x^l_m \xrightarrow{\mathbf{R}} \sum_{m'} D^l_{mm'}(\mathbf{R}) x^l_{m'}$, where the equivariant feature vector $x^l_m$ carries the irreducible representation of the SO(3) group of dimension $2l+1$, $l$ is an integer or half-integer, and $m$ is an integer or half-integer varying between $-l$ and $l$. Moreover, feature vectors can be converted to equivariant tensors (or vice versa) via the Wigner–Eckart theorem: $l_1 \otimes l_2 = |l_1 - l_2| \oplus \cdots \oplus (l_1 + l_2)$, which changes tensor product '$\otimes$' into direct sum '$\oplus$'. For example, a representation of $l_1 \otimes l_2$ is constructed by $\mathbf{X}^{l_1 l_2}_{m_1 m_2} = \sum_{l_3, m_3} C^{l_1 l_2 l_3}_{m_1 m_2 m_3} x^{l_3}_{m_3}$, where $C^{l_1 l_2 l_3}_{m_1 m_2 m_3}$ are Clebsch–Gordan coefficients. The equivariant rotation transformation rule of the tensor is $\mathbf{X}^{l_1 l_2}_{m_1, m_2} \xrightarrow{\mathbf{R}} \sum_{m_1', m_2'} D^{l_1}_{m_1, m_1'}(\mathbf{R}) D^{l_2}_{m_2, m_2'}(\mathbf{R}) \mathbf{X}^{l_1 l_2}_{m_1', m_2'}$. When excluding the spin degrees of freedom and selecting real-valued $D^l(\mathbf{R})$ for the orbital space, $\mathbf{h}^{l_1 l_2}_{m_1, m_2}$ follows the same transformation rule as $\mathbf{X}^{l_1 l_2}_{m_1, m_2}$ (equation (1)). Thus, the spin-unpolarized DFT Hamiltonian can be represented by the tensor, making the rotation-equivariant property preserved. In contrast, using ENNs to construct the spin–orbital DFT Hamiltonian is more complicated for the following reasons. First, the DFT Hamiltonian matrix subblock corresponds to a special representation of $\left(l_1 \otimes \frac{1}{2}\right) \otimes \left(l_2^* \otimes \frac{1}{2}^*\right)$, where $l^*$ denotes a representation whose rotation transformation matrix is the complex conjugate of that for the representation $l$. Second, complex-valued quantities are generally involved in half-integer representations, but only real-valued ENNs are currently available for practical computation. The problem is addressed in the DeepH-E3 method[11], which proposes to convert the representation $l^*$ into $l$ by unitary transformation, and reduce half-integer representations into integer ones by applying the Wigner–Eckart theorem.

## Neural network model

On notations of building blocks in Fig. 2, 'E3Linear' is defined as

$$\text{E3Linear}(\mathbf{x})^l_{cm} = \sum_{c'} W^l_{cc'} x^l_{c'm} + b^l_c, \tag{3}$$

where $l$ is the angular momentum quantum number, $m$ is the magnetic quantum number, $c$ denotes the channel index, and $W^l_{cc'}$ and $b^l_c$ are learnable weights. To preserve equivariance requirements, biases $b^l_c \neq 0$ only for equivariant features $\mathbf{x}$ with $l = 0$, even spatial-inversion index and even time-reversal index.

'$(U\mathbf{x}) \otimes (V\mathbf{y})$' denotes the tensor product operation between features $\mathbf{x}$ and $\mathbf{y}$, where $U$ and $V$ are learnable parameters. 'E3Layer-Norm' is used to normalize the feature while preserving equivariance[11]:

$$\text{E3LayerNorm}(\mathbf{x})^l_{cm} = g^l_c \frac{x^l_{cm} - \mu^l_m}{\sigma^l + \epsilon} + h^l_c, \tag{4}$$

where $\mu^l_m$ and $\sigma^l$ are the mean and the standard deviation of features, respectively, $g^l_c$ and $h^l_c$ are learnable weights, and $\epsilon$ is denominator for realizing numerical stability. $h^l_c$ has the same equivariance requirements as $b^l_c$ in equation (3).

The layer of 'Non-linearity' produces the non-linearity activation function or the scalar gate on features $\mathbf{x}$ with $l = 0$ or $l > 0$, respectively. For $l = 0$ features with even spatial-inversion index and even time-reversal index, the activation function sigmoid linear unit is used. In addition, for other $l = 0$ features, the odd activation function tanh is used to preserve equivariance with respect to spatial inversion and time reversal.

'Embedding' is used to construct initial features and transform the interatomic distance vector $\mathbf{r}_{ij}$ and magnetic moment $\mathbf{m}_i$ into equivariant vectors. Initial vertex features $\mathbf{v}^{(0)}_i$ are vector embeddings of nuclear charge $Z_i$. Initial edge features $\mathbf{e}^{(0)}_{ij}$ are interatomic distances $|\mathbf{r}_{ij}|$ expanded by Gaussian functions. Real spherical harmonics $\mathbf{Y}_{lm}$ acting on the vector input $\mathbf{r}_{ij}$ and $\mathbf{m}_i$ are used to introduce equivariant features with arbitrary $l$, and update vertex and edge features. Inputs $\mathbf{Y}_{lm}(\mathbf{m}_i)$ for non-magnetic atoms $i$ are set to zero.

In the magnetic moment layer, 'MLP' is multilayer perceptron, and 'Aggregation' is used to aggregate the magnetic moment information in the $R_C$ range to the edge feature $\mathbf{e}^l_{ij}$:

$$S^l_{ik} = \text{MLP}^l_{\text{left}}(\mathbf{e}^{\text{scalar}}_{ik}), k \in \mathcal{N}_i \tag{5}$$

$$W^l_{jk'} = \text{MLP}^l_{\text{right}}(\mathbf{e}^{\text{scalar}}_{jk'}), k' \in \mathcal{N}_j \tag{6}$$

$$\text{Aggregation}^l(S^l, W^l, \mathbf{r}, \mathbf{m})$$
$$= \left( \sum_{k \in \mathcal{N}_i} S^l_{ik} \left( \mathbf{Y}^l(\hat{r}_{ik}) \| \mathbf{Y}^l(\hat{m}_i) \| \mathbf{Y}^l(\hat{m}_k) \right) \right) \bigg\|$$
$$\left( \sum_{k' \in \mathcal{N}_j} W^l_{jk'} \left( \mathbf{Y}^l(\hat{r}_{jk'}) \| \mathbf{Y}^l(\hat{m}_j) \| \mathbf{Y}^l(\hat{m}_{k'}) \right) \right), \tag{7}$$

where 'scalar' means spatial-inversion-even and time-reversal-even equivariant feature with $l = 0$, $\mathcal{N}_i$ is the set of neighbors of atom $i$, $S$ and $W$ are the outputs of equations (5) and (6), $\hat{r}$ and $\hat{m}$ are unit vectors of $\mathbf{r}$ and $\mathbf{m}$, respectively, and '$\|$' denotes vector concatenation. Equation (7) introduces strict locality in the same spirit as ref. 15.

## Details of neural network training

The xDeepH model was implemented with PyTorch, PyTorch Geometric and e3nn[23] libraries. The initial vertex and edge features were 64-dimensional. For the vertex and edge layers, equivariant vertex and edge features were set to be 64 × 0eE + 32 × 1oE + 16 × 1eE + 16 × 2eE + 8 × 3o E + 8 × 4eE, where 64 × 0eE denotes 64 time-reversal-even equivariant vectors carrying the $l = 0$ representation with even parity, and 32 × 0oE denotes 32 time-reversal-even equivariant vectors carrying the $l = 1$ representation with odd parity. For the intermediate magnetic moment layer, equivariant edge features were set to be 64 × 0eE + 32 × 1oE + 16 × 1eE + 16 × 2eE + 8 × 3oE + 8 × 4eE + 64 × 0eO + 32 × 1oO + 16 × 1eO + 16 × 2eO + 8 × 3oO + 8 × 4eO, where 64 × 0eO denotes 64 time-reversal-odd equivariant vectors carrying the $l = 0$ representation with even parity. The output features of the last magnetic moment layer were set to carry the irreducible representation required to construct the target Hamiltonian matrix. Finally, the final edge features were passed through a

E3Linear layer to obtain the Hamiltonian matrix block. $l = 0$ to $l = 4$ was used for $\mathbf{Y}_{lm}(\mathbf{r}_{ij})$ and $\mathbf{Y}_{lm}(\mathbf{m}_i)$. The neural network we use included three vertex layers, three edge layers and three magnetic moment layers.

Training was performed on an NVIDIA RTX 3090 graphics processing unit with a batch size of 1. The initial learning rate was set to be 0.002, which decreases by a factor of 0.5 when the loss does not decrease after 120 epochs. Supplementary Table 3 summarizes the number of parameters of the neural network models and the data splitting used for training, validation and testing, respectively, in each dataset. For $NiBr_2$, we found that the deep learning model without E3LayerNorm can achieve slightly lower loss. As a result, we used the model without E3LayerNorm for this dataset, which brings a small difference in the number of parameters.

### Computational details

DFT calculations were performed by the OpenMX software package (version 3.9)[24] using pseudo-atomic localized basis functions, norm-conserving pseudopotentials and the Perdew–Berke–Ernzerhof-type exchange-correlation functional. Constraint DFT as implemented in OpenMX is applied to study systems with specified magnetic configurations. In all calculations, we constrain the orientation of magnetic atoms (Ni and Cr) with 0.5 eV as the prefactor of spin constraint. The spin–orbit coupling is included in the DFT calculations. For $NiBr_2$, the basis sets of Ni6.0H-$s3p2d1$ and Br7.0-$s3p2d2$ are used, including 14 basis functions for Ni with cut-off radius $r_c = 6.0$ bohr and 19 basis functions for Br with $r_c = 7.0$ bohr. The DFT + $U$ method with a Hubbard correction of $U = 4.0$ eV is applied to describe the $3d$ orbitals of Ni. For $CrI_3$, the basis sets of Cr6.0-$s3p2d1$ and I7.0-$s3p2d2$ pseudo-atomic orbitals are used, including 14 basis functions for Cr with $r_c = 6.0$ bohr and 19 basis functions for I with $r_c = 7.0$ bohr. The convergence of basis sets in the DFT calculations is confirmed by test calculations (Supplementary Section 6.1). The influence of basis-set size on the training efficiency of xDeepH is also checked (Supplementary Section 6.2). The energy cut-off is set to be 300 rydberg. To generate datasets, a Monkhorst–Pack $k$-mesh of $7 \times 7 \times 1$ is used for supercell calculations of monolayer $NiBr_2$ with 27 atoms, monolayer $CrI_3$ with 32 atoms and bilayer $CrI_3$ with 64 atoms. Meanwhile, a Monkhorst–Pack $k$-mesh of $3 \times 15 \times 1$ is used for monolayer $NiBr_2$ with $19 \times 1$ spiral magnetism (114 atoms), $1 \times 1 \times 13$ for (16, 16) $CrI_3$ nanotubes (256 atoms) and $3 \times 5 \times 1$ for the $2 \times 1$ supercell of twisted bilayer $CrI_3$ with twist angle 81.79° (224 atoms).

### Dataset preparation

The datasets in this work involve systems with randomly perturbed atomic and magnetic configurations. Magnetic configurations are constrained by specifying orientations of local magnetic moments of magnetic sites (Ni or Cr). For uniform sampling of magnetic moment orientations, we first generate $M_x, M_y, M_z \approx N(0, 1)$ for each magnetic site, where $N(0, 1)$ stands for standard normal distribution. They are then normalized by dividing $\sqrt{M_x^2 + M_y^2 + M_z^2}$. The resulting distribution of magnetic orientation vector is equivalent to uniform distribution on the unit sphere.

For monolayer $NiBr_2$, we prepare DFT datasets by calculating 500 $3 \times 3$ supercells of $NiBr_2$ with 27 atoms at the equilibrium lattice structure with random magnetic moment orientations. For monolayer $CrI_3$, 100 different atomic structures of $2 \times 2$ supercells with 32 atoms are prepared by introducing random atomic displacements (up to 0.1 Å) to the equilibrium lattice structure. For each atomic structure, 10 random magnetic configurations are generated with random magnetic moment orientations. In total, 1,000 supercell structures with random atomic and magnetic configurations are included in the datasets of monolayer $CrI_3$. Training sets of bilayer $CrI_3$ are composed of $2 \times 2$ supercells containing 64 atoms. To simulate a local environment of different interlayer stacking patterns, the second layer is arranged with an overall shift with respect to the first layer. The overall shift is sampled by a uniform $16 \times 16$ grid of the supercell, yielding 256 atomic

configurations. In addition, random displacements up to 0.1 Å are introduced to each atom about their equilibrium positions. For each atomic configuration, 10 random magnetic configurations are generated with the aforementioned method. In total, 2,560 bilayer $CrI_3$ supercells with 256 unique atomic configurations and 2,560 unique magnetic configurations are prepared. All datasets are randomly divided into training, validation and test sets with a ratio of 6:2:2.

## Data availability

Source data for Extended Data Figs. 1 and 2 is available with this paper. The dataset used to train the deep learning model is available at Zenodo[29].

## Code availability

The code used in the current study is available at GitHub (https://github.com/mzjb/xDeepH) and Zenodo[30].

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

## Acknowledgements
We thank F. Zheng for providing the structures of magnetic skyrmions. This work was supported by the Basic Science Center Project of NSFC (Grant No. 52388201), the National Science Fund for Distinguished Young Scholars (Grant No. 12025405), the National Natural Science Foundation of China (Grant No. 11874035), the Ministry of Science and Technology of China (Grant Nos. 2018YFA0307100 and 2018YFA0305603), the Beijing Advanced Innovation Center for Future Chip (ICFC) and the Beijing Advanced Innovation Center for Materials Genome Engineering.

## Author contributions
Y.X. and W.D. proposed the project and supervised H.L. and Z.T. in carrying out the research, with the help of X.G. and N.Z. All authors discussed the results. Y.X. and H.L. prepared the paper with input from the other co-authors.

## Competing interests
The authors declare no competing interests.

## Additional information

**Correspondence and requests for materials** should be addressed to Wenhui Duan or Yong Xu.

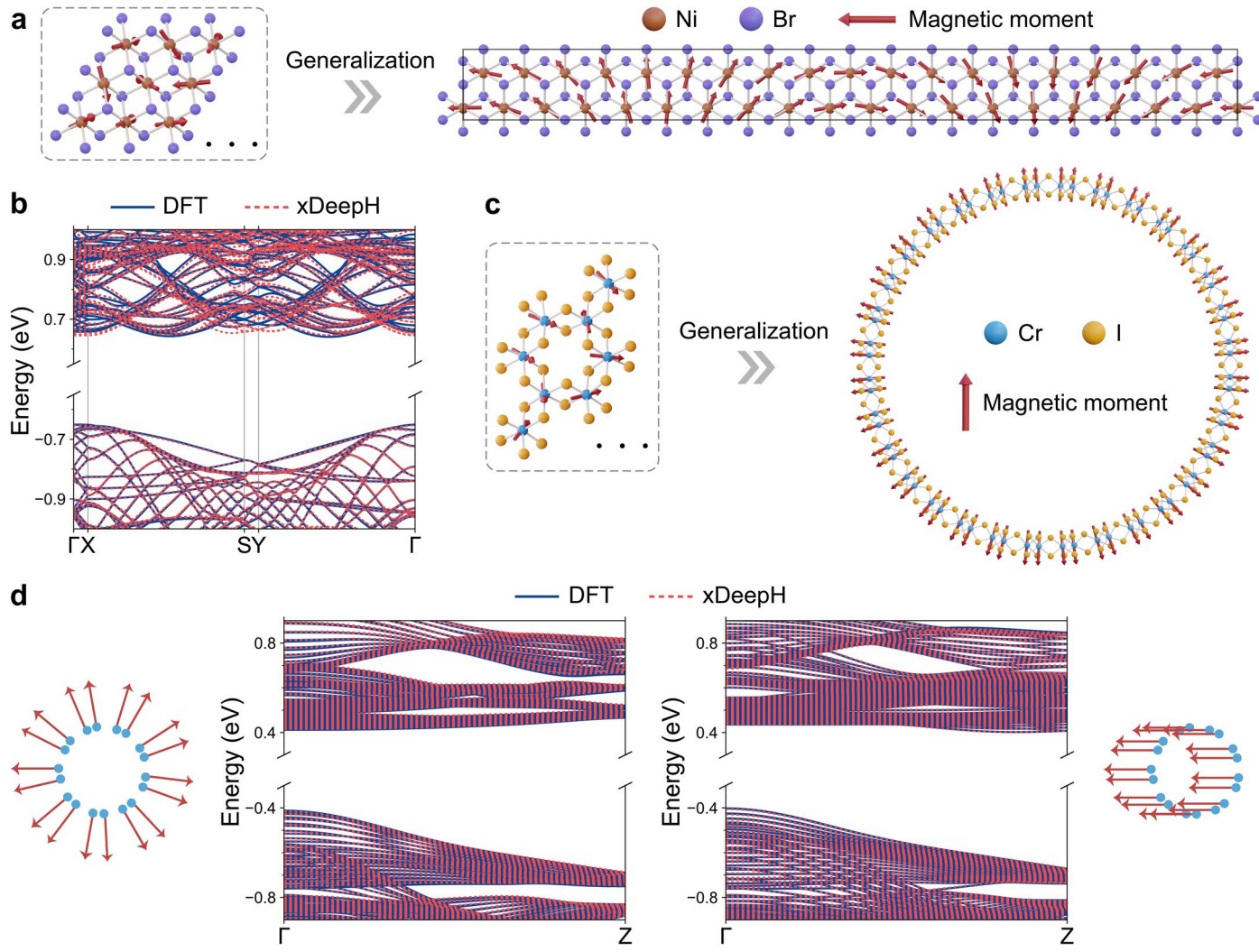

**Extended Data Fig. 1 | Application of xDeepH to study spin-spiral and nanotube magnets. a**, Example study on NiBr$_2$, which uses DFT data on monolayer structures with random magnetic configurations for training and predicts on spiral magnetic structures. **b**, Band structures of monolayer NiBr$_2$ with 19 × 1 spiral magnetism computed by DFT and xDeepH. **c**, Example study on CrI$_3$, which uses DFT data on monolayer structures for training and predicts on nanotubes. **d**, Band structures of (16, 16) CrI$_3$ nanotube with the non-collinear magnetization (left) and collinear ferromagnetism (right) computed by DFT and xDeepH. Magnetic moments are denoted by arrows. Γ, X, S, Y, and Z denote different high-symmetry $k$-points of the Brillouin zone.

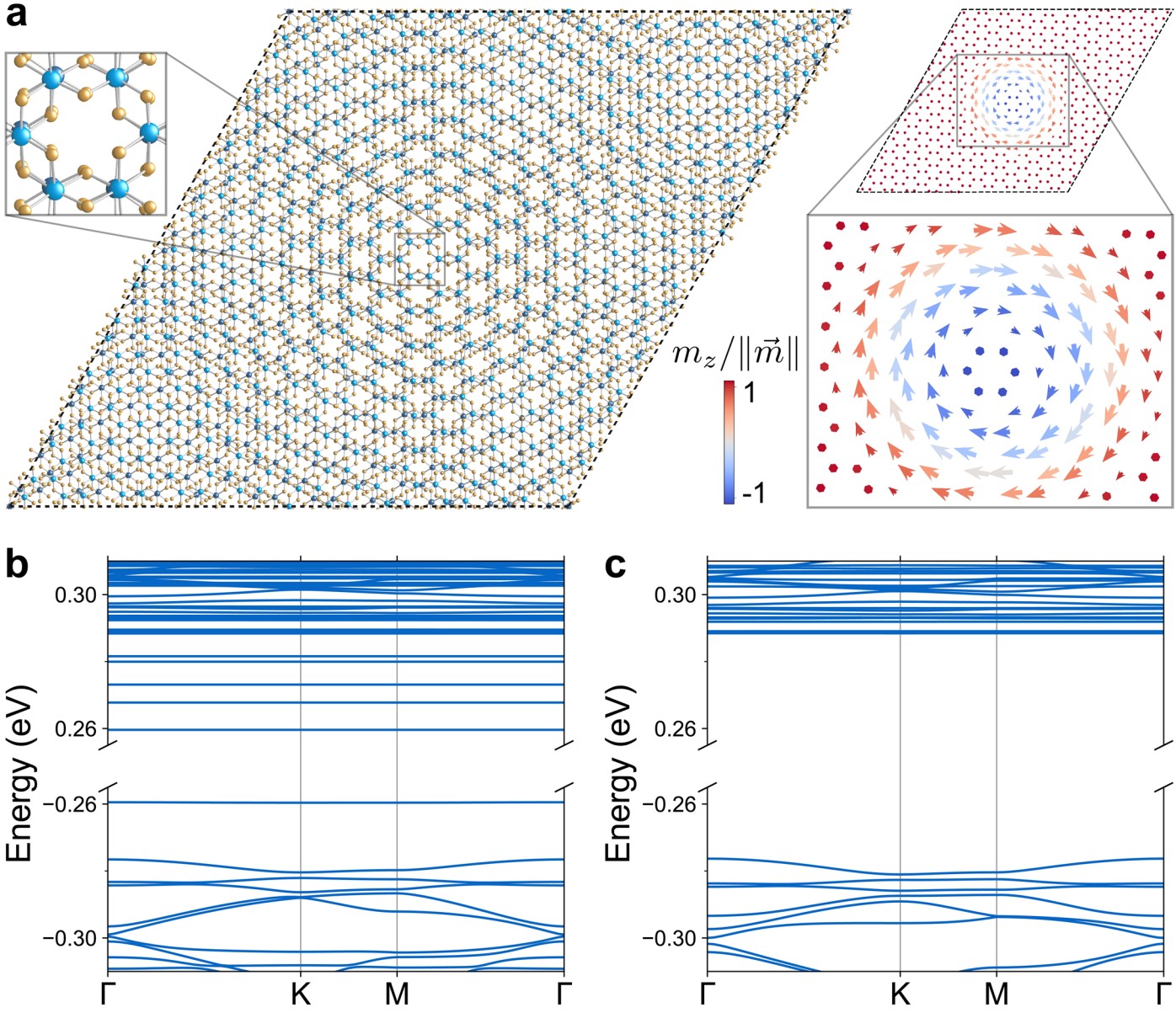

**Extended Data Fig. 2 | Application of xDeepH to study Moiré magnets without or with magnetic skyrmion. a**, Schematic atomic structure and magnetic skyrmion in the Moiré-twisted bilayer $CrI_3$ (twist angle 63.48° and 4,336 atoms per supercell)[28]. Magnetic moments of the top $CrI_3$ layer are labeled by colored arrows, whose out-of-plane components are shown by the color. The underlying $CrI_3$ layer is in the ferromagnetic configuration with up magnetic moments. **b,c**, Band structures of the Moiré-twisted bilayer $CrI_3$ (**b**) in the ferromagnetic configuration and (**c**) in the magnetic skyrmion configuration. Γ, K and M denote different high-symmetry $k$-points of the Brillouin zone.

