## [Peer Review File · Nature Computational Science]

Peer Review Information

Journal: Nature Computational Science

Manuscript Title: Deep-learning electronic-structure calculation of magnetic superstructures

Corresponding author name(s): Wenhui Duan, Yong Xu

Reviewer Comments & Decisions:

Decision Letter, initial version:

Dear Professor Xu,

Your manuscript "Deep-learning electronic-structure calculation of magnetic superstructures" has now been seen by 3 referees, whose comments are appended below. You will see that while they find your work of interest, they have raised points that need to be addressed before we can make a decision on publication.

The referees' reports seem to be quite clear. Naturally, we will need you to address all of the points raised.

While we ask you to address all of the points raised, the following points need to be substantially worked on:

- Please provide additional comparisons to SpookyNet.
- Please provide additional demonstrations/comparisons to DFT methods to demonstrate the efficiency claims.
- Please be sure that all code and data is up to date as a result of the changes made during the revision process. Please also be sure to provide a clear README file with instructions.

Please use the following link to submit your revised manuscript and a point-by-point response to the referees' comments (which should be in a separate document to any cover letter):

[REDACTED]

** This url links to your confidential homepage and associated information about manuscripts you may have submitted or be reviewing for us. If you wish to forward this e-mail to co-authors, please delete this link to your homepage first. **

To aid in the review process, we would appreciate it if you could also provide a copy of your manuscript files that indicates your revisions by making use of Track Changes or similar mark-up tools. Please also ensure that all correspondence is marked with your Nature Computational Science

reference number in the subject line.

In addition, please make sure to upload a Word Document or LaTeX version of your text, to assist us in the editorial stage.

If you have any issues when updating your Code Ocean capsule during the revision process, please email the Code Ocean support team [Cc'ing me](mailto:codeocean@springernature.com).

To improve transparency in authorship, we request that all authors identified as 'corresponding author' on published papers create and link their Open Researcher and Contributor Identifier (ORCID) with their account on the Manuscript Tracking System (MTS), prior to acceptance. ORCID helps the scientific community achieve unambiguous attribution of all scholarly contributions. You can create and link your ORCID from the home page of the MTS by clicking on 'Modify my Springer Nature account'. For more information please visit www.springernature.com/orcid.

We hope to receive your revised paper within three weeks. If you cannot send it within this time, please let us know.

Best regards,

Kaitlin McCardle
Editor
Nature Computational Science

Reviewers comments:

Reviewer #1 (Remarks to the Author):

In this work, the authors propose a deep-learning-based method for the electronic structure calculation of magnetic superstructures. The deep learning model not only needs to respect the Euclidean symmetry, as many models for atomic properties have to, but it also needs to respect the time-reversal symmetry. The methodology here is novel, and the numerical studies are solid. As such, in my view, this work is a timely contribution to the field and will have broad applications. I recommend an acceptance after minor revision.

I just have several concerns about the efficiency and scalability of the method. It is unclear to me how much the quantitative improvement of the current implementation of the method is upon DFT methods. For their applications, could the authors plot the CPU/GPU wall time VS system size? This will be beneficial for practitioners who want to apply this method to problems they are interested in. In addition, it seems that the method is sensitive to the basis one uses in the DFT method. For the applications presented in this work, have the authors checked the convergence of the setup for the basis set in their DFT calculations? This may be presented in Supplementary Materials (and maybe also the efficiency of the method VS the size of the basis set.)

Reviewer #2 (Remarks to the Author):

The authors presented a manuscript entitled "Deep-learning electronic-structure calculation of magnetic structures". Overall, the manuscript is well-structured and the main conclusions are consistent with the results. They have successfully developed an equivariant neural network architecture (xDeepH) to learn the dependence of spin-orbital DFT Hamiltonians on atomic and magnetic structures. The implementation of the architecture and the physical/mathematical models involved on it are well-explained in the manuscript. The case studies were carefully selected to demonstrated the robustness of the ENN. I am positively surprised by the results that the authors have shown and discussed, however, before acceptance for publication, there are some major corrections that the authors must address:

-In 2021, it was published a different ENN which also uses spin features as input parameters, this is SpookyNet [O.T. Unke et al., Nature Communications 12, 7273, (2021)]. The authors of that work didn't carry out case studies similar to these ones presented in this manuscript but it would be interesting to compare the performance of both ENN, if possible.

-It is well-known that NN models are faster than DFT calculations in computing physicochemical properties, however, when the NN architecture gets more complex (e.g., equivariant features combined with physical models), the number of parameters of the trained NN model also largely increases. This, in the majority of cases, slows down the computation of properties. Can the authors show a comparison of simulation time for the case studies they have performed? How many parameters do their ENN models have?

-Are the computed magnetic properties depending on the optimized geometry? This could be a problem for the applicability of the trained ENN models, since the user will have to always run geometry optimisation using the same level of theory as the data used to train the models. Is it possible to implement the learning of atomic forces and energies in this architecture to perform geometry optimisations and then magnetic property calculations?

-Since the authors are predicting magnetic properties of periodic systems, is it possible to obtain spin-up and spin-down band structures (and from them the corresponding density of states) in collinear magnetic systems? This is a widely performed analysis in the study of magnetic properties.

-Can the authors show evidences of the following statement "In general, qualities with local property are more favorable for deep learning than nonlocal ones."? Page 2. Perhaps, citing some references.

-Analyzing the results for the first case study (monolayer NiBr₂), one can see that valence bands are accurately computed by the trained ENN model in comparison with conduction bands. Could the authors explain from where are these discrepancies originated? It would be good to compare density of states as well. Similar effect is observed for the third case study, see Figure S7.

-Can the authors briefly explain how the electrical susceptibility was computed in the SI?

-For the generation of the dataset in the third case study, are different angle orientations considered in the 256 atomic configurations? The authors should clarify this to have a better understanding of the performance of the trained ENN model.

-Is xDeepH only limited to periodic systems? I know that equivariant networks have a good

performance for highly symmetric systems. Therefore, I would like to see some case studies in open systems like organometallic magnets. This would show the broad range of applicability of the architecture.

-Explanation of how to train models is also missing in the SI.

Reviewer #2 (Remarks on code availability):

I was able to access the code on the website codeocean.com but I couldn't run the script "run". Moreover, the download option was not activated. It would be better if the authors create a Github repository that the reviewers could access without any problem. The README file with further details about the code or how to train a model is also missing.

Reviewer #3 (Remarks to the Author):

The manuscript reports about a deep learning method for calculations of magnetic properties of complex materials. I found the manuscript to be interesting and can recommend it for publication in Nature Computational science, when the authors properly reply my following questions and modify the manuscript accordantly.

1. The method is supposed to be suitable for large systems. First, small (initial) system should be calculated with a conventional DFT method and then by deep learning a large system can be studied. Let's suppose one has to calculate a magnetic alloy. the source of magnetic moments are magnetic impurities. How can the suggested approach be applied for alloys? The impurity might be distributed randomly or can be clustered. The results will be strongly dependent on the impurity distribution. So the main question is how to account for short-range effects.

2. I'm curious about that physical quantities which are sensitive to the dimensionality. Let's suppose the initial cells (3D) have a special magnetisation direction. How can the code account for a possible change of the magnetisation direction in a film with a surface or in a heterostructure. Such a change can not be learned from supercell and can be related to a particular dimensionality or to a particular termination.

3. In the discussion the authors claim that the method can be easily extended for other applications mentioning electron-magnon, magnon-phonon interaction etc. Can the authors provide more details? The mentioned problems are used to be solved with a linear response approach, for that a full susceptibility must be calculated. In this case, a Dyson equation should be solved (not a Hamiltonian problem). I'm curious whether the method can be extended for a linear response approach.

Author Rebuttal to initial comments

Reviewer #1:

In this work, the authors propose a deep-learning-based method for the electronic structure calculation of magnetic superstructures. The deep learning model not only needs to respect the Euclidean

symmetry, as many models for atomic properties have to, but it also needs to respect the time-reversal symmetry. The methodology here is novel, and the numerical studies are solid. As such, in my view, this work is a timely contribution to the field and will have broad applications. I recommend an acceptance after minor revision.

Response: We gratefully thank the referee for his/her careful review on our manuscript. We also sincerely appreciate the referee for the insightful suggestions and high recognition on our work. Below we give a point-to-point response to the comments raised.

Comment 1: I just have several concerns about the efficiency and scalability of the method. It is unclear to me how much the quantitative improvement of the current implementation of the method is upon DFT methods. For their applications, could the authors plot the CPU/GPU wall time VS system size? This will be beneficial for practitioners who want to apply this method to problems they are interested in.

Response: We thank the referee for the helpful suggestion. To make a quantitative comparison, we selected $N \times 1$ supercells ($N = 1, 2, 4, 8 \dots$) of ferromagnetic monolayer CrI_3 for example studies, computed the DFT Hamiltonian matrices by DFT self-consistent calculations and xDeepH using the same CPU node, and plotted the CPU wall time versus system size in Fig. R1. The efficiency advantage of xDeepH over DFT is significant and becomes more considerable as the system size increases. Note that xDeepH would work much more efficiently if using GPU nodes and its computational efficiency could be further improved.

Figure R1. Computation time to construct the DFT Hamiltonian matrices of ferromagnetic monolayer CrI_3 supercells with varying system sizes by DFT self-consistent calculations versus by xDeepH. All the

calculations were performed by one compute node equipped with two AMD EPYC 7542 CPUs, although xDeepH would work much more efficiently on GPU nodes.

As a response, we added Supplementary Section 5 “Computational cost comparison between xDeepH and DFT” in the Supplementary Information, presented the above discussion in it, and included Fig. R1 as Supplementary Figure 10.

Comment 2: In addition, it seems that the method is sensitive to the basis one uses in the DFT method. For the applications presented in this work, have the authors checked the convergence of the setup for the basis set in their DFT calculations? This may be presented in Supplementary Materials (and maybe also the efficiency of the method VS the size of the basis set.)

Response: The referee pointed out an important issue on the convergence of basis set in DFT and xDeepH calculations. For xDeepH, the DFT Hamiltonian as a function of atomic and magnetic structures is learned by neural networks. The object of deep learning is to reproduce DFT calculation results as accurate as possible. As systematically demonstrated in this work, the xDeepH method can do this job well for varying kinds of material systems. The use of a larger basis set would introduce larger DFT Hamiltonian matrices for deep learning. This would moderately increase the training time and slightly increase the inference time, keeping the computational complexity unchanged. In this sense, the method itself is independent of the use of different basis sets for DFT calculations.

On the other hand, the DFT results are basis dependent, so are the calculation results of xDeepH. For DFT calculations, we have carefully checked the influence of basis set on band-structure calculations and confirmed the convergence of basis set before doing routine calculations. Test results on monolayer NiBr₂, monolayer and bilayer CrI₃ are shown in Fig. R2. Further increasing the size of basis set has minor influence on the calculated band structures, implying that the basis set used in our DFT calculations is large enough to ensure convergence.

Figure R2. Band structures of (a) monolayer NiBr_2 , (b) monolayer CrI_3 , (c) bilayer CrI_3 with $C2/m$ interlayer stacking, and (d) bilayer CrI_3 with $R3$, interlayer stacking calculated by using different basis sets. The material systems are in the ferromagnetic state for (a, b, d) and in the A-type antiferromagnetic state for (c). “Basis 1” denotes $\text{Ni}6.0\text{H-}s3p2d1$, $\text{Br}7.0\text{-}s3p2d2$, $\text{Cr}6.0\text{-}s3p2d1$, and $\text{I}7.0\text{-}s3p2d2$ for Ni, Br, Cr, and I, respectively. “Basis 2” denotes $\text{Ni}6.0\text{H-}s3p2d2f1$, $\text{Br}7.0\text{-}s3p2d2f1$, $\text{Cr}6.0\text{-}s3p2d2f1$, and $\text{I}7.0s3p3d2f1$ for Ni, Br, Cr, and I, respectively. “Basis 1” is used in our routine DFT calculations. “Basis 2” is larger in size than “Basis 1”.

Following the referee’s suggestion, we also tested the training efficiency of xDeepH versus the size of basis set (Table R1). The number of model parameters and the time-consuming per iteration of training do not increase much for increasing the size of basis set.

Table R1. Training efficiency of xDeepH versus the size of basis set. The material system used for test is the ferromagnetic monolayer CrI_3 2×2 supercell with 32 atoms. The neural-network training is performed on an NVIDIA RTX 3090 GPU with a batch size of 1. The basis set used in our routine DFT calculations is highlighted in bold.

Basis functions	Number of parameters	Training time per iteration (seconds)
Cr6.0-$s3p2d1$; I7.0-$s3p2d2$	2126632	1.6
Cr6.0- $s3p2d1$; I7.0- $s3p2d2f1$	2530330	1.8
Cr6.0- $s3p2d2f1$; I7.0- $s3p3d2f1$	2903748	1.9

As a response, we added Supplementary Section 6 “Basis set tests” in the Supplementary Information, presented the above discussion in it, and included Fig. R2 as Supplementary Figure 11 and Table R1 as Supplementary Table 2.

Reviewer #2:

The authors presented a manuscript entitled “Deep-learning electronic-structure calculation of magnetic structures”. Overall, the manuscript is well-structured and the main conclusions are consistent with the results. They have successfully developed an equivariant neural network architecture (xDeepH) to learn the dependence of spin-orbital DFT Hamiltonians on atomic and magnetic structures. The implementation of the architecture and the physical/mathematical models involved on it are well explained in the manuscript. The case studies were carefully selected to demonstrate the robustness of the ENN. I am positively surprised by the results that the authors have shown and discussed, however, before acceptance for publication, there are some major corrections that the authors must address:

Response: We gratefully thank the referee for his/her careful review on our manuscript. We also sincerely appreciate the referee for the insightful suggestions and high recognition on our work. Below we give a point-to-point response to the comments raised.

Comment 1: In 2021, it was published a different ENN which also uses spin features as input parameters, this is SpookyNet [O.T. Unke et al., Nature Communications 12, 7273, (2021)]. The authors of that work didn’t carry out case studies similar to these ones presented in this manuscript but it would be interesting to compare the performance of both ENN, if possible.

Response: We thank the referee for introducing the work of SpookyNet, which was originally cited as Ref. [9] in our manuscript. Both SpookyNet and xDeepH are developed to provide a complete description of material properties, which not only consider the nuclear degrees of freedom as input, but also take the electronic degrees of freedom into account. Both methods are based on equivariant neural network (ENN) models. It is thus meaningful to compare the two methods, as suggested by the referee.

Figure R3. Workflow and application of deep-learning models, including SpookyNet and xDeepH, which are used to study (a) atomic structure and (b) electronic structure, respectively. The atomic coordinate ($\{\vec{r}_i\}$) and nuclear charge ($\{Z_i\}$) are inputs to both ENNs. The input of SpookyNet also includes two integer number, the total charge (Q) and spin (S) of the molecule. In contrast, xDeepH considers magnetic moments of individual atoms ($\{\vec{m}_i\}$) as a vector input. SpookyNet predicts the total energy (E), and xDeepH predicts the DFT Hamiltonian matrix (H).

The workflow and application of SpookyNet and xDeepH are illustrated and compared in Fig. R3. SpookyNet is a deep neural network used to predict the total energy for a given molecular geometry and specified electronic degrees of freedom including the total charge and spin of the molecule. In contrast, xDeepH is a deep neural network used to predict the DFT Hamiltonian for given atomic and magnetic configurations of crystalline materials. Their deep-learning objects and target material systems are distinct. Moreover, the two methods have different applications: The former is applied to study atomic structure, and the latter is used to study electronic structure. In this context, the performance of the two ENN methods cannot be directly and quantitatively compared.

Nevertheless, it is worthwhile to compare the neural network architectures of the two ENNs. SpookyNet takes the total spin of the system into account via a scalar input (i.e., a positive integer number). This network does not need to handle rotational and time-reversal equivariance with respect to spin features. Moreover, SpookyNet predicts the total energy, which is invariant under Euclidean symmetry. SpookyNet projects rotationally equivariant features to invariant ones for improving computational efficiency. In contrast, xDeepH includes magnetic features of individual atoms as a vector input, and predicts the DFT Hamiltonian matrix that is more difficult to treat than the scalar output of SpookyNet. All the input, internal, and output feature vectors of xDeepH are kept equivariant (not simply invariant). Thus, xDeepH can explicitly preserve the rotational and time-reversal equivariance with respect to magnetic features, which is critical for achieving good performance. In brief, the two ENNs have their own advantages: The ENN of SpookyNet is computationally more efficient for predicting invariant physical quantities, and the ENN of xDeepH is more desirable for predicting equivariant physical quantities.

As a response, we added a few sentences on page 5: “A previous work developed a deep neural network SpookeyNet to learn the total energy and atomic forces of molecules, which not only considers the nuclear degrees of freedom as input, but also takes the electronic degrees of freedom (including the total charge and spin) into account [9]. It is also based on the ENN method. A comprehensive comparison between xDeepH and SpookeyNet is presented in Supplementary Section 7.” Moreover, we added Supplementary Section 7 “Comparison between xDeepH and SpookeyNet” in the Supplementary Information, presented the comprehensive comparison between xDeepH and SpookeyNet in it, and included Fig. R3 as Supplementary Figure 12.

Comment 2: It is well-known that NN models are faster than DFT calculations in computing physicochemical properties, however, when the NN architecture gets more complex (e.g., equivariant features combined with physical models), the number of parameters of the trained NN model also largely increases. This, in the majority of cases, slows down the computation of properties. Can the authors show a comparison of simulation time for the case studies they have performed? How many parameters do their ENN models have?

Response: We thank the referee for the helpful suggestion. To make a quantitative comparison, we selected $N \times 1$ supercells ($N = 1, 2, 4, 8 \dots$) of ferromagnetic monolayer CrI₃ for example studies, computed the DFT Hamiltonian matrices by DFT self-consistent calculations and xDeepH using the same CPU node, and plotted the CPU wall time versus system size in Fig. R4. The efficiency advantage of xDeepH over DFT is significant and becomes more considerable as the system size increases. Note that xDeepH would work much more efficiently if using GPU nodes and its computational efficiency could be further improved.

Following the referee’s suggestion, we summarized the number of parameters of ENN models for studies of different material systems in Supplementary Table 3.

Figure R4. Computation time to construct the DFT Hamiltonian matrices of ferromagnetic monolayer CrI₃ supercells with varying system sizes by DFT self-consistent calculations versus by xDeepH. All the calculations were performed by one compute node equipped with two AMD EPYC 7542 CPUs, although xDeepH would work much more efficiently on GPU nodes.

As a response, we added Supplementary Section 5 “Computational cost comparison between xDeepH and DFT” in the Supplementary Information, presented the above discussion in it, included Fig. R4 as Supplementary Figure 10, and added Supplementary Table 3 to present the number of parameters of ENN models.

Comment 3: Are the computed magnetic properties depending on the optimized geometry? This could be a problem for the applicability of the trained ENN models, since the user will have to always run geometry optimisation using the same level of theory as the data used to train the models. Is it possible to implement the learning of atomic forces and energies in this architecture to perform geometry optimisations and then magnetic property calculations?

Response: The computed magnetic properties are dependent on the optimized geometry. As pointed out by the referee, the optimized geometry is an input of xDeepH, but cannot be obtained from xDeepH in its present design. Thus, neural-network material calculations demand the training of extra neural network models for geometry optimization. As illustrated in Fig. R3, xDeepH solves the problem of predicting the DFT Hamiltonian for a given geometric ($\{\mathcal{R}\}$) and magnetic ($\{\mathcal{M}\}$) configuration. Developing deep-learning models to predict the total energy is another important issue, which can be used for performing geometry optimization efficiently. Since the ENN of xDeepH can be used to study equivariant physical quantities and the total energy is a special kind of equivariant quantities, the neural network architecture of xDeepH could be adapted to learn the total energy. Special effort, however, is

required to optimize the neural network performance. We would like to explore this topic in future works.

Comment 4: Since the authors are predicting magnetic properties of periodic systems, is it possible to obtain spin-up and spin-down band structures (and from them the corresponding density of states) in collinear magnetic systems? This is a widely performed analysis in the study of magnetic properties.

Response: The xDeepH method is designed to learn non-collinear DFT calculation results of magnetic systems. It is naturally applicable to learn collinear DFT calculation results, considering that the latter corresponds to a special, simplified case of the former. On the other hand, for the non-collinear DFT calculations, one may project the Bloch eigenstates onto the spin-up and spin-down subspaces, and obtain spin-up and spin-down projected band structure or density of states. This is demonstrated by analyzing the non-collinear DFT calculation results of ferromagnetic monolayer CrI₃ (Fig. R5).

Figure R5. Spin-up (colored blue) and spin-down (colored red) projected band structure (the left panel) and density of states (the right panel) of ferromagnetic monolayer CrI₃ obtained from the non-collinear DFT calculation.

Comment 5: Can the authors show evidences of the following statement “In general, qualities with local property are more favorable for deep learning than nonlocal ones.”? Page 2. Perhaps, citing some references.

Response: We thank the referee for the helpful suggestion. To overcome the formidable computational cost of large-scale *ab initio* calculations, deep learning methods are developed to train neural network

models with *ab initio* data of small-scale systems and apply the trained neural networks to predict material properties of large-scale systems. It is quite challenging to train neural network models for an accurate description of non-local properties of large-scale materials, because such kind of material properties complicatedly depend on the global chemical environment. The argument is supported by references. For instance, Ref. [O. T. Unke *et al.*, Chem. Rev. **121**, 10142 (2021)] discussed the challenges faced by neural network force fields when non-local effects are not negligible. Ref. [Y. Chen *et al.*, J. Chem. Theory Comput. **17**, 170 (2021)] and Ref. [L. Zepeda-Núñez *et al.*, J. Comput. phys. **443**, 110523 (2021)] emphasized the importance of locality to achieve good generalization ability and computational efficiency of deep learning models.

As a response, we cited the above references as Refs. [24-26] in the revised manuscript.

Comment 6: Analyzing the results for the first case study (monolayer NiBr₂), one can see that valence bands are accurately computed by the trained ENN model in comparison with conduction bands. Could the authors explain from where are these discrepancies originated? It would be good to compare density of states as well. Similar effect is observed for the third case study, see Figure S7.

Response: We thank the referee for pointing out this issue. To analyze the discrepancy between valence and conduction bands, we performed an orbital analysis on the band structure of ferromagnetic monolayer NiBr₂ (Fig. R6). The results indicate that the valence and conduction bands are mainly contributed by *p*-orbitals of Br atoms and *d*-orbitals of Ni atoms, respectively. Due to the different orbital contribution, electronic states of conduction bands are more sensitive to the change of magnetic configuration. The strongly correlated nature of electronic states would make an accurate prediction of DFT Hamiltonian challenging. This could explain the discrepancy between valence and conduction bands of the system.

Figure R6. Orbital projected band structure of ferromagnetic monolayer NiBr₂. Contributions of Ni *d*-orbital and Br *p*-orbital are colored blue and red, respectively.

Following the referee's suggestion, we also compared the density of states for monolayer NiBr₂ supercells calculated by DFT and xDeepH and appended the results to the corresponding band structures

(Fig. R7). The different performance on describing valence and conduction bands looks more obvious.

Figure R7. Band structures (the left panel) and density of states (the right panel) of the three typical tests samples of monolayer NiBr₂ with (a) the best, (b) median and (c) worst MAE of the DFT Hamiltonian matrix element computed by DFT and xDeepH.

In contrast, we did not observe similar discrepancy for CrI₃ (see Fig. 3d), although the material also has *p*-orbital valence bands and *d*-orbital conduction bands. The underlying reason, however, is unclear, which demands in-depth research. Figure S7 (Supplementary Figure 8 in the revised manuscript) gives a comparison of DFT calculated band structures of CrI₃ nanotube with two different magnetic configurations, showing the influence of magnetic configuration on band structure. The results indicate that the conduction bands are more sensitive to the change of magnetic configuration than the valence bands, in consistent with the judgement on basis of orbital analysis.

As a response, we added the above discussion in Supplementary Section 4.1, included Fig. R6 as Supplementary Figure 4, and appended the comparison of density of states for monolayer NiBr₂ in Supplementary Figure 3.

Comment 7: Can the authors briefly explain how the electrical susceptibility was computed in the SI?

Response: The electrical susceptibility was computed by the formula:

$$\chi''_{ab} = \frac{e^2}{\epsilon_0} \frac{1}{4\pi\hbar} \sum_{\mathbf{k}} \frac{r_{ab}(\mathbf{k})}{\omega_{\mathbf{k}} - \omega - i\eta}, \quad (1)$$

where a, b and c are cartesian directions, ϵ_0 is the vacuum permittivity, \hbar is the reduced Planck's constant, e is the charge of electron, and η is an infinitesimal broadening factor. $\omega_{\mathbf{k}} = (E_{\mathbf{k}} - E_{\mathbf{k}'})/\hbar$ and $f_{\mathbf{k}} = f_{\mathbf{k}'} - f_{\mathbf{k}}$ are the difference of band energy eigenvalues and Fermi-Dirac occupations of bands n and m at wave vector \mathbf{k} , respectively. $r_{ab}(\mathbf{k})$ is the Berry connection when $n \neq m$ or zero when $n = m$. The quantity is calculated by the method developed in Ref. [New J. Phys.

21, 093001 (2019)] as implemented by the code HopTB.jl (<https://github.com/HopTB/HopTB.jl>).

As a response, we added the above description in Supplementary Section 4.3.

Comment 8: For the generation of the dataset in the third case study, are different angle orientations considered in the 256 atomic configurations? The authors should clarify this to have a better understanding of the performance of the trained ENN model.

Response: Different angle orientations were considered in the 256 atomic configurations. In total, bilayer CrI₃ supercells with 256 unique atomic configurations and 2560 unique magnetic configurations were included in the dataset.

As a response, we clarify the number of unique magnetic configurations to avoid misunderstanding in the section "Dataset preparation".

Comment 9: Is xDeepH only limited to periodic systems? I know that equivariant networks have a good performance for highly symmetric systems. Therefore, I would like to see some case studies in open systems like organometallic magnets. This would show the broad range of applicability of the architecture.

Response: The DFT Hamiltonian under localized basis is learned and predicted by xDeepH, which can be used to perform electronic structure calculation for both periodic and non-periodic systems. Thus, xDeepH is not limited to periodic systems.

We agree with the referee that ENNs have good performance for highly symmetric systems. The symmetry-related data could be aggregated together by equivariant transformations, making the deep learning more efficient and accurate. The advantage of ENN is obvious in the study of crystalline materials, as demonstrated by xDeepH. For the study of open systems, the benefit of using ENNs becomes less, and more training data would be demanded to achieve comparable prediction accuracy. As suggested by the referee, open systems like organometallic magnets could be used to study spin-dependent electronic properties. However, xDeepH is originally designed and optimized to study crystalline materials. The neural network models should be reoptimized for the study of molecular systems, which is tedious and time consuming. Moreover, organometallic magnets are physically distinct from solid magnets. The former system usually only has a few magnetic sites that are nearly isolated from each other. Physically, if the magnetic sites are far away from each other, changing their relative spin orientations would essentially not affect the electron density and band structure (or energy levels of molecules). In addition, if the spin-orbit coupling is negligible, the spin and orbital subspaces are decoupled, and then a global rotation of spin configuration would not affect the band structure. Therefore, the change of magnetic orientation typically has minor influence on electronic structure in organometallic magnets. To some extent, the study of organometallic magnets departs from the research interest of the current work, which focuses on magnetic superstructures and related physics. Hopefully the referee would agree that this is not an immediate and simple task and it would be better to do that in other works. Nevertheless, the work suggested by the referee is valuable from the perspective of deep learning, which points out a new direction for the development of xDeepH.

Comment 10: Explanation of how to train models is also missing in the SI.

Response: Following the referee's suggestion, we added Supplementary Section 8 "Details of Neural network training" in the Supplementary Information, presented more complete details about neural network training in it, and moved the section "Details on training" of the main text into it.

Comment 11: I was able to access the code on the website codeocean.com but I couldn't run the script "run". Moreover, the download option was not activated. It would be better if the authors create a Github repository that the reviewers could access without any problem. The README file with further details about the code or how to train a model is also missing.

Response: It might be caused by the unstable network connection with CodeOcean, since sometimes one have to click the "Reproducible Run" button several times to run successfully. In addition, there will be no response for "run" when one runs out of the compute time on CodeOcean (10 hours/month). After clicking the "Reproducible Run" button, CodeOcean will first install the environment (about ten minutes) and then run the code.

Since the download option is not activated in CodeOcean, we would like to submit our code and the README file with further details alongside the revised manuscript. A GitHub repository will be created if the current work is accepted for publication.

Reviewer #3:

The manuscript reports about a deep learning method for calculations of magnetic properties of complex materials. I found the manuscript to be interesting and can recommend it for publication in Nature Computational science, when the authors properly reply my following questions and modify the manuscript accordantly.

Response: We gratefully thank the referee for the careful review and constructive suggestions. We also sincerely appreciate the referee for the high recognition of our work. Below we will make a point-topoint response to the points raised.

Comment 1: The method is supposed to be suitable for large systems. First, small (initial) system should be calculated with a conventional DFT method and then by deep learning a large system can be studied. Let's suppose one has to calculate a magnetic alloy. the source of magnetic moments are magnetic impurities. How can the suggested approach be applied for alloys? The impurity might be distributed randomly or can be clustered. The results will be strongly dependent on the impurity distribution. So the main question is how to account for short-range effects.

Response: We thank the referee for the helpful comment. Magnetic alloys with magnetic moments contributed by impurities can be studied by xDeepH using the following procedures:

1) Prepare DFT datasets:

First, a suitable size of supercell is selected, which should be not too small for avoiding artificial interactions between periodic images but not too large for making the ab initio calculations affordable. Then, magnetic impurities are introduced into the supercells with varying impurity concentrations, random impurity distributions, and random magnetic orientations. If the magnetic impurities are prone to be clustered, supercells with clustered impurities should also be carefully considered. The guiding principle is that the generated samples should include diverse kinds of local chemical environments, especially those close to in realistic material systems. Finally, DFT calculations are performed to calculate the Hamiltonians of these supercells with random atomic and magnetic structures for the generation of datasets.

2) Train neural network models:

Use part of datasets for neural-network training. Test the performance of neural networks with the remaining datasets.

3) Predict electronic properties of new structures:

Give a new structure unseen in the DFT datasets. The trained neural networks of xDeepH will be used to predict its DFT Hamiltonian. Then all the electronic properties of this structure in the singleparticle picture could be calculated from the predicted DFT Hamiltonian.

As emphasized in the manuscript, xDeepH utilizes the nearsightedness property of electronic matter in order to make predictions on large structures after being trained on small structures, as long as the local chemical environments of the large structures are similar to those of the small structures in the training set. Provided that this principle is followed in the design of the training set, xDeepH could well describe the dependence of ab initio tight-binding Hamiltonian on the local chemical environment, making accurate prediction of electronic structure feasible.

As a response, we added the above description in Supplementary Section 9.1.

Comment 2: I'm curious about that physical quantities which are sensitive to the dimensionality. Let's suppose the initial cells (3D) have a special magnetisation direction. How can the code account for a possible change of the magnetisation direction in a film with a surface or in a heterostructure. Such a change can not be learned from supercell and can be related to a particular dimensionality or to a particular termination.

Response: We thank the referee for the helpful comment. The xDeepH method does not have the ability to learn from 3D bulk structures and make predictions on quasi-2D structures, such as surfaces, interfaces or thin films. For studying physical quantities that are sensitive to the dimensionality, the training set should include sample structures of different dimensions. These samples should contain diverse local chemical environments, especially those close to environments in the new material structure to be studied. By using such kind of training datasets, xDeepH could be applied to study electronic properties of new structures.

The referee mentioned that the magnetic easy axis may be different in the bulk, thin films, or heterostructures. In fact, the most stable magnetic orientation is determined by the total energy. To study this property, one may apply neural network models to learn the dependence of total energy on

atomic and magnetic structures. Then, one can obtain the optimized atomic and magnetic structures by minimizing the total energy. Using the stable atomic and magnetic structures as the input of xDeepH, one may predict the DFT Hamiltonian and the electronic properties of the specified material.

Comment 3: In the discussion the authors claim that the method can be easily extended for other applications mentioning electron-magnon, magnon-phonon interaction etc. Can the authors provide more details? The mentioned problems are used to be solved with a linear response approach, for that a full susceptibility must be calculated. In this case, a Dyson equation should be solved (not a Hamiltonian problem). I'm curious whether the method can be extended for a linear response approach.

Response: We thank the referee for the helpful comment. Theoretically, with the knowledge about the dependence of total energy and effective electronic Hamiltonian (like DFT Hamiltonian) on the atomic structure, one may study material properties related to phonons, electrons, and electron-phonon interactions [G. D. Mahan, *Many-Particle Physics*, (Springer Science, 2000)]. Similarly, with the knowledge about the dependence of total energy and DFT Hamiltonian on the magnetic structure, one may investigate material properties related to magnons, electron-magnon interactions and phononmagnon interactions. Since the dependence of total energy and DFT Hamiltonian on the atomic and magnetic structures can be learned by neural network models, one may use the trained neural network models to very efficiently derive the matrix elements describing electron-magnon coupling, phononmagnon coupling, etc. In contrast, traditional methods for the calculation of these matrix elements (e.g., ab initio linear response theory) are considerably time consuming. Once the matrix elements are obtained, the Dyson equation is solved to study physical properties, such as the lifetimes of electrons and magnons. The major bottleneck for ab initio linear response theory calculations is to obtain these matrix elements instead of solving the Dyson equation. Intriguingly, the derivative calculations of matrix elements can be efficiently computed by the automatic differentiation technique of neural networks. Therefore, it is highly possible that the xDeepH method will outperform the computationally expensive traditional methods of ab initio linear response theory.

As a response, we added the above discussion in Supplementary Section 9.2.

Summary of changes (Revisions are colored blue in the revised manuscript.)

Main text:

- On page 5, a few sentences were added in the first and third paragraphs.
- On page 6, one sentence was added in the Discussion part.
- On pages 6-8, three sentences were added in the Methods part.
- References [24-26] were added.

Supplementary Information:

- Two paragraphs were added in Supplementary Section 4.
- Supplementary Figure 3 was updated.
- Supplementary Figures 4, 10, 11, and 12 were added.
- Supplementary Tables 2 and 3 were added.
- Supplementary Sections 5-9 were added.

Decision Letter, first revision:

14th February 2023

Dear Dr. Xu,

Thank you for submitting your revised manuscript "Deep-learning electronic-structure calculation of magnetic superstructures" (NATCOMPUTSCI-22-1288A). It has now been seen by the original referees and their comments are below. The reviewers find that the paper has improved in revision, and therefore we'll be happy in principle to publish it in Nature Computational Science, pending minor revisions to satisfy the referees' final requests and to comply with our editorial and formatting guidelines.

TRANSPARENT PEER REVIEW

Nature Computational Science offers a transparent peer review option for original research manuscripts. We encourage increased transparency in peer review by publishing the reviewer comments, author rebuttal letters and editorial decision letters if the authors agree. Such peer review material is made available as a supplementary peer review file. **Please state in the cover letter 'I wish to participate in transparent peer review' if you want to opt in, or 'I do not wish to participate in transparent peer review' if you don't.** Failure to state your preference will result in delays in accepting your manuscript for publication.

Thank you again for your interest in Nature Computational Science Please do not hesitate to contact me if you have any questions.

Sincerely,

Kaitlin McCardle
Editor
Nature Computational Science

ORCID

Reviewer #1 (Remarks to the Author):

The authors have addressed my comments properly and I recommended the paper to be published as is.

Reviewer #2 (Remarks to the Author):

The authors have addressed all my comments with clarity and in details, improving the quality of the manuscript. I recommend the revised version of the manuscript for publication after the authors address this minor correction.

Not further revision is needed from my side.

-The authors should add few sentences about the discussion of spin-up/spin-down calculations and the application of their method to non-periodic systems in the section "Discussion" of the main text.

Reviewer #2 (Remarks on code availability):

The code was installed properly and the training started with the dataset suggested by the authors. README file has enough instructions to install and use the code.

Reviewer #3 (Remarks to the Author):

I'm happy with the revised version of the manuscript and recommend it for publication in Nature computational science.

Author rebuttal, second revision:

Reviewer #2:

The authors have addressed all my comments with clarity and in details, improving the quality of the manuscript. I recommend the revised version of the manuscript for publication after the authors address this minor correction. Not further revision is needed from my side.

Response: We gratefully thank the referee for his/her supportive comments. In the following we will make a point-to-point response to the comments raised.

The authors should add few sentences about the discussion of spin-up/spin-down calculations and the application of their method to non-periodic systems in the section "Discussion" of the main text.

Response: Thank you for your valuable comment. As suggested by the referee, we added two subsections to present the corresponding discussions, including Supplementary Section 9.3 titled "Study of collinear magnetic systems" and Supplementary Section 9.4 titled "Study of non-periodic systems". Moreover, we added the original Fig. R5 as Supplementary Figure 13, and mentioned the above discussions in the "Discussion" part.

The code was installed properly and the training started with the dataset suggested by the authors. README file has enough instructions to install and use the code.

Response: Thank you for informing us about code availability. We will be delighted to receive any feedbacks of the code.

Final Decision Letter:

Dear Professor Xu,

We are pleased to inform you that your Brief Communication "Deep-learning electronic-structure calculation of magnetic superstructures" has now been accepted for publication in Nature Computational Science.

Once your manuscript is typeset, you will receive an email with a link to choose the appropriate publishing options for your paper and our Author Services team will be in touch regarding any additional information that may be required.

Please note that *Nature Computational Science* is a Transformative Journal (TJ). Authors may publish their research with us through the traditional subscription access route or make their paper immediately open access through payment of an article-processing charge (APC). Authors will not be required to make a final decision about access to their article until it has been accepted. [Find out more about Transformative Journals](https://www.springernature.com/gp/open-research/transformative-journals)

Authors may need to take specific actions to achieve [compliance with funder and institutional open access mandates](https://www.springernature.com/gp/open-research/funding/policy-compliance-faqs). If your research is supported by a funder that requires immediate open access (e.g. according to [Plan S principles](https://www.springernature.com/gp/open-research/plan-s-compliance)) then you should select the gold OA route, and we will direct you to the compliant route where possible. For authors selecting the subscription publication route, the journal's standard licensing terms will need to be accepted, including [self-archiving policies](https://www.springernature.com/gp/open-research/policies/journal-policies). Those licensing terms will supersede any other terms that the author or any third party may assert apply to any version of the manuscript.

Acceptance of your manuscript is conditional on all authors' agreement with our publication policies (see <https://www.nature.com/natcomputsci/for-authors>). In particular your manuscript must not be published elsewhere and there must be no announcement of the work to any media outlet until the publication date (the day on which it is uploaded onto our web site).

Before your manuscript is typeset, we will edit the text to ensure it is intelligible to our wide readership and conforms to house style. We look particularly carefully at the titles of all papers to ensure that they are relatively brief and understandable.

Once your manuscript is typeset and you have completed the appropriate grant of rights, you will

receive a link to your electronic proof via email with a request to make any corrections within 48 hours. If, when you receive your proof, you cannot meet this deadline, please inform us at rjsproduction@springernature.com immediately.

If you have queries at any point during the production process then please contact the production team at rjsproduction@springernature.com. Once your paper has been scheduled for online publication, the Nature press office will be in touch to confirm the details.

Content is published online weekly on Mondays and Thursdays, and the embargo is set at 16:00 London time (GMT)/11:00 am US Eastern time (EST) on the day of publication. If you need to know the exact publication date or when the news embargo will be lifted, please contact our press office after you have submitted your proof corrections. Now is the time to inform your Public Relations or Press Office about your paper, as they might be interested in promoting its publication. This will allow them time to prepare an accurate and satisfactory press release. Include your manuscript tracking number NATCOMPUTSCI-22-1288B and the name of the journal, which they will need when they contact our office.

About one week before your paper is published online, we shall be distributing a press release to news organizations worldwide, which may include details of your work. We are happy for your institution or funding agency to prepare its own press release, but it must mention the embargo date and Nature Computational Science. Our Press Office will contact you closer to the time of publication, but if you or your Press Office have any inquiries in the meantime, please contact press@nature.com.

We welcome the submission of potential cover material (including a short caption of around 40 words) related to your manuscript; suggestions should be sent to Nature Computational Science as electronic files (the image should be 300 dpi at 210 x 297 mm in either TIFF or JPEG format). We also welcome suggestions for the Hero Image, which appears at the top of our <http://www.nature.com/natcomputsci> home page; these should be 72 dpi at 1400 x 400 pixels in JPEG format. Please note that such pictures should be selected more for their aesthetic appeal than for their scientific content, and that colour images work better than black and white or grayscale images. Please do not try to design a cover with the Nature Computational Science logo etc., and please do not submit composites of images related to your work. I am sure you will understand that we cannot make any promise as to whether any of your suggestions might be selected for the cover of the journal.

To assist our authors in disseminating their research to the broader community, our SharedIt initiative provides you with a unique shareable link that will allow anyone (with or without a subscription) to read the published article. Recipients of the link with a subscription will also be able to download and

print the PDF.

Best regards,

Kaitlin McCardle
Editor
Nature Computational Science

P.S. Click on the following link if you would like to recommend Nature Computational Science to your librarian: https://www.springernature.com/gp/librarians/recommend-to-your-library

** Visit the Springer Nature Editorial and Publishing website at www.springernature.com/editorial-and-publishing-jobs for more information about our career opportunities. If you have any questions please click here. **